# Uncertainty-aware retinal layer segmentation in OCT through probabilistic signed distance functions

**Mohammd Mohaiminul Islam**[1,2]                          M.M.ISLAM@UVA.NL
**Coen de Vente**[1,2]                                      C.W.DEVENTE@UVA.NL
**Bart Liefers**[3]                                         B.LIEFERS@ERASMUSMC.NL
**Caroline Klaver**[3]                                      C.C.W.KLAVER@ERASMUSMC.NL
**Erik J Bekkers**[1]                                       E.J.BEKKERS@UVA.NL
**Clara I. Sánchez**[1,2]                                   C.I.SANCHEZGUTIERREZ@UVA.NL

[1] *Informatics Institute, University of Amsterdam, Netherlands.*

[2] *Department of Biomedical Engineering and Physics, Amsterdam UMC, Netherlands.*

[3] *Department of Ophthalmology & Epidemiology, Erasmus MC, Rotterdam, Netherlands.*

**Editors:** Accepted for publication at MIDL 2024

## Abstract

In this paper, we present a new approach for uncertainty-aware retinal layer segmentation in Optical Coherence Tomography (OCT) scans using probabilistic signed distance functions (SDF). Traditional pixel-wise and regression-based methods primarily encounter difficulties in precise segmentation and lack of geometrical grounding respectively. To address these shortcomings, our methodology refines the segmentation by predicting a signed distance function (SDF) that effectively parameterizes the retinal layer shape via level set. We further enhance the framework by integrating probabilistic modeling, applying Gaussian distributions to encapsulate the uncertainty in the shape parameterization. This ensures a robust representation of the retinal layer morphology even in the presence of ambiguous input, imaging noise, and unreliable segmentations. Both quantitative and qualitative evaluations demonstrate superior performance when compared to other methods. Additionally, we conducted experiments on artificially distorted datasets with various noise types—shadowing, blinking, speckle, and motion—common in OCT scans to showcase the effectiveness of our uncertainty estimation. Our findings demonstrate the possibility to obtain reliable segmentation of retinal layers, as well as an initial step towards the characterization of layer integrity, a key biomarker for disease progression. Our code is available at https://github.com/niazoys/RLS_PSDF.

**Keywords:** Probabilistic signed distance function, Implicit representation, OCT

## 1. Introduction

In recent years, deep learning methods have demonstrated remarkable success in automating the delineation and segmentation of retinal layers in Optical Coherence Tomography (OCT) scans. The task of retinal layer segmentation is primarily tackled via pixel-wise classification, an approach extensively explored in recent works (Li et al., 2020; Liu et al.,

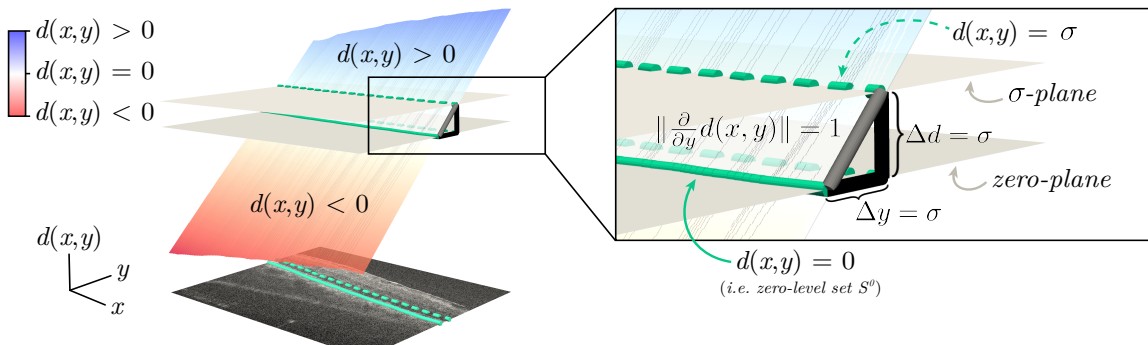

Figure 1: Left: The zero level set (green) of a signed distance function (SDF) $d$ parametrizes a retinal layer in a OCT B-scan. Right: Due to the Eikonal constraint $\|\frac{\partial}{\partial y}d(x,y)\| = 1$, uncertainty in the SDF $d$, here represented as displacement $\Delta d = \sigma$ in the level, induces an equal displacement $\Delta y$ of the contour.

2024; Kugelman et al., 2022; Sousa et al., 2021; Lou et al., 2020; Chen et al., 2019; Kugelman et al., 2018; Santos et al., 2019). This technique, however, is known to have challenges in precisely delineating the small structures and nuanced contours when employing cross-entropy or dice loss (Abraham and Khan, 2019). The issue is further exacerbated when the layer targeted for segmentation is exceptionally thin, often reduced to a one-pixel width line for layer boundaries or as a consequence of disease manifestation, resulting in inaccurate (Lin et al., 2023), and disconnected boundaries (see Fig. 6) (Lan et al., 2020; Sousa et al., 2021).

The alternative is to switch to a regression-based approach, predicting layer coordinates per vertical scan (A-scan) line, which addresses the problem of disconnected layer boundaries. Notably, deep learning models that take a two-dimensional B-scan as input and directly output a coordinate for each A-scan (Liefers et al., 2019; Ngo et al., 2019), and models that directly predict the height of each layer, ensuring a correct layer ordering (Morelle et al., 2023), have been proposed. However, the regression-based approach lacks mechanistic interpretability as the predictions are not in one-to-one correspondence from input to output, i.e. it reduces all pixel information in a vertical column to a single coordinate, making the relationship between input and anticipated boundary ambiguous. Due to this reduction, the geometric grounding of the predictions is challenged, limiting the neural network's ability to exploit geometric priors and regularities.

We present a new level set approach that overcomes both the challenges of thin layer segmentation and the lack of geometric grounding. Our approach involves predicting an SDF that parametrizes the layer boundary via a level set. The predicted SDF promotes an enhanced spatial congruence between the input and output via a one-to-one pixel correspondence, thereby delivering more robust supervisory signals for the learning algorithm. Additionally, our approach is amenable to probabilistic modeling through a Gaussian likelihood assumption.

The SDF has seen application in broader medical image segmentation contexts (Xue et al., 2020; Raju et al., 2022; Bogensperger et al., 2023), but their utility in retinal layer segmentation remains unexplored. On the other hand, the literature on aleatoric uncertainty estimation via Gaussian likelihood in medical image segmentation is extensive (Abdar et al., 2021; Gawlikowski et al., 2021). Although (Dong et al., 2018) introduced a probabilistic framework for SDF using a Normal-Beta mixture distribution, their approach does not extend to the uncertainty in segmentation or shape inference challenges. To the best of our knowledge, this research represents the initial effort to employ SDF to address the challenge of segmenting thin retinal layer boundaries and incorporate Gaussian uncertainty in SDF for segmentation tasks.

Following are the key contributions of this work:

- We present a redefined approach to thin layer boundary segmentation with SDF.
- We extend Gaussian modeling for aleatoric uncertainty estimation in our framework.
- We show that the proposed uncertainty estimation can offer a means to flag unreliable segmentation, particularly under compromised layer visibility or integrity.

## 2. Method

Consider an OCT B-scan, or image, $I : \mathcal{X} \times \mathcal{Y} \to \mathbb{R}$ that assigns to each coordinate $x, y$ an intensity value $I(x, y)$, with $x \in \mathcal{X}$ the horizontal coordinate and $y \in \mathcal{Y}$ the vertical coordinate. Let $a_x \in I$ denote a single A-scan at horizontal location $x$ as $a_x : \mathcal{Y} \to \mathbb{R}$, and it is defined $a_x(y) := I(x, y)$. The ground truth of a retinal layer can be parametrized as a function $y^{gt} : \mathcal{X} \to \mathcal{Y}$ that assigns to every a-scan a corresponding vertical $y$ coordinate. The $y$ location in single A-scan $a_x$ may then be denoted as $y_x^{gt} := y^{gt}(x)$. With this notation in place, we present three methods for retinal layer segmentation: pixel-wise, regression, and signed distance function (SDF) approach.

**Pixel-wise approach**: A neural network $f$ predicts a segmentation function $s : \mathcal{X} \times \mathcal{Y} \to [0, 1]$ that returns a probability for the layer to each position. Specifically, a network with parameters $\theta$ generates $s = f^{segm}[I; \theta]$, such that ideally $s(x, y_x^{gt}) = 1$ at the location of the layer, and zero otherwise. Typically, the pixel-wise segmentation predictions are parametrized with Bernoulli distributions, giving a probability for object presence and uncertainty that is directly reflected by the entropy (i.e., $-s \log(s) - (1 - s) \log(1 - s)$) of the distribution. Notably, this approach can only express uncertainty in the presence of the structure, but not its shape.

**Regression approach ($REGR$ & $pREGR$)**: A neural network $f^{regr}$ predicts a regression function $y^{regr} : \mathcal{X} \to \mathcal{Y}$ that parametrizes the layer similarly as done in the ground truth, generating $y^{regr} = f^{regr}[I; \theta]$ such that $\forall_x : y_x^{regr} \approx y_x^{gt}$. Such a model will be referred to with label $REGR$. To incorporate uncertainty in the predicted location we let the neural network parametrize a predictive distribution

$$p(y(x) \mid I, \theta) = \mathcal{N}(\mu(x), \sigma(x)^2), \tag{1}$$

which we model as a Gaussian distribution with mean $\mu(x) = f_\mu^{regr}[I;\theta](x)$ and variance $\sigma(x) = f_\sigma^{regr}[I;\theta](x)$. Such a probabilistic approach will be referred to with $pREGR$.

**Signed distance function approach ($SDF$)**: A neural network $f^{sdf}$ predicts a signed distance function $d : \mathcal{X} \times \mathcal{Y} \to \mathbb{R}$ with the property that it satisfies the Eikonal constraint, that is $\forall_{x,y} : \|\frac{\partial}{\partial y} d(x,y)\| = 1$, and $d(x, y_x^{gt}) = 0$, such that $d$ can be interpreted as a distance to the layer. The Eikonal constraint ensures that $d$ linearly increases when moving away from the boundary. Note that in our method the Eikonal constraint is enforced by explicit construction of the ground truth to train the model. Then from a predicted SDF $d$, the actual location of the layer is obtained by solving

$$d(x, y) = 0\,, \tag{2}$$

i.e., by finding the zero-level set of $d$, where the level set of a function $d$ is defined as the set of coordinates $S^l$ for which it equals the given level $l$. In our setting, this means that $S^l := \{(x, y) \in X \times Y \mid d(x, y) = l\}$. Since we can define the SDF per A-scan as $d_x$ we can also define the per-A-scan level set as $S_x^l := \{y \in Y \mid d_x(y) = l\}$.

Due to the laminar structure of the retina, and the Eikonal constraint, we can assume that the per-A-scan zero level set consists of only a single element, which we denote with $y^{sdf}(x)$ and which is the unique solution of (2). We thus have $S_x^0 = \{y^{sdf}(x)\}$. In contrast to the regression method which *explicitly* (directly) parametrizes a layer through a contour function $y^{regr} : \mathcal{X} \to \mathcal{Y}$, the signed distance function approach *implicitly* parametrizes $y^{sdf} : \mathcal{X} \to \mathcal{Y}$ as the zero level set of $d$. We label the latter method with *SDF*.

**Probabilistic SDF-based regression ($pSDF$)**: We model uncertainty on the location of the layer by letting the neural network predict the expected distance $\mu(x, y) = f_\mu^{sdf}[I;\theta](x, y)$ as well as an uncertainty for that prediction $\sigma(x, y) = f_\sigma^{sdf}[I;\theta](x, y)$, to parametrize a predictive distribution for the SDF $d$ via

$$p(d(x, y) \mid I, \theta) = \mathcal{N}(\mu(x, y), \sigma(x, y)^2)\,. \tag{3}$$

The uncertainty in the predicted SDF value is directly related to uncertainty in the actual contour location. Namely, if the SDF is normally distributed as above, then the uncertainty $\sigma(x, y)$ on the SDF value translates to uncertainty on the vertical location via

$$p(y(x) \mid I, \theta) = \mathcal{N}(y^{sdf}(x), \sigma(x, y^{sdf}(x))^2)\,, \tag{4}$$

where $y^{sdf}(x)$ is the solution of (2) at $x$. This result follows from the fact that by the Eikonal constraint, we have $d(x, y + \sigma) = d(x, y) + \sigma$, see Fig. 1. We label this probabilistic SDF-based regression approach with $pSDF$.

**Optimization objective**: With these probabilistic models in place, we take as optimization objective the likelihood of the ground truth under the probabilistic models. For the *SDF* approach, we construct the ground-truth $d^{gt}(x, y)$ following the process detailed

in A.1. The negative log-likelihood (NLL) for it at a given pixel $(x, y)$ is obtained by

$$\ln p(d^{gt}(x,y) \mid I, \theta) = -\frac{1}{2} \left[ \frac{(d^{gt}(x,y) - f^{sdf}_\mu[I;\theta](x,y))^2}{f^{sdf}_\sigma[I;\theta]^2(x,y)} + \ln f^{sdf}_\sigma[I;\theta]^2(x,y) + \ln(2\pi) \right], \quad (5)$$

which under i.i.d. assumption leads to the overall loss (summed over all pixels):

$$\mathcal{L}(\theta) = -\sum_{x,y} \ln p(d^{gt}(x,y) \mid I, \theta). \quad (6)$$

The NLL becomes a least squares loss if $\sigma = 1$, which we use for *REGR* and *SDF*.

Table 1: Layer segmentation performance (measured with mean absolute error (MAE) in pixels).

| Method | Layer | MAE (internal) | MAE (external) |
|---|---|---|---|
| REGR | ILM | $0.97_{\pm 0.72}$ | $2.37_{\pm 1.83}$ |
| | RPE | $1.32_{\pm 0.51}$ | $2.54_{\pm 2.14}$ |
| | BM | $1.76_{\pm 1.48}$ | $5.28_{\pm 2.94}$ |
| | Avg. | 1.35 | 3.40 |
| pREGR | ILM | $0.85_{\pm 0.71}$ | $2.98_{\pm 2.41}$ |
| | RPE | $1.28_{\pm 0.82}$ | $2.71_{\pm 2.00}$ |
| | BM | $1.80_{\pm 1.50}$ | $3.69_{\pm 1.78}$ |
| | Avg. | 1.31 | 3.12 |
| SDF (Our) | ILM | $0.33_{\pm 0.42}$ | $1.61_{\pm 0.75}$ |
| | RPE | $0.60_{\pm 0.41}$ | $1.02_{\pm 1.64}$ |
| | BM | $0.79_{\pm 0.42}$ | $1.39_{\pm 1.01}$ |
| | Avg. | 0.57 | 1.34 |
| pSDF (Our) | ILM | $0.32_{\pm 0.44}$ | $1.63_{\pm 0.78}$ |
| | RPE | $0.59_{\pm 0.40}$ | $1.02_{\pm 1.47}$ |
| | BM | $0.75_{\pm 0.29}$ | $1.27_{\pm 0.82}$ |
| | Avg. | **0.55** | **1.30** |

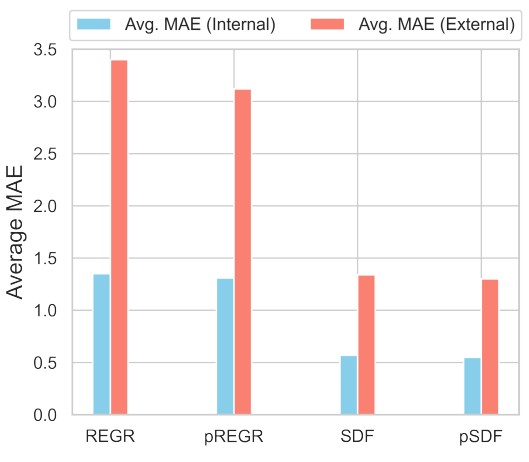

Figure 2: Layer segmentation performance summary on *internal* and *external* test set.

## 3. Experimental Setup

**Datasets.** We utilized the public dataset *(internal)* from (Farsiu et al., 2014), comprising 384 OCT volumes from AMD and control participants, with three manually annotated retinal layers: the inner limiting membrane (ILM), the retinal pigment epithelium (RPE), and Bruch's membrane (BM). OCT volumes were acquired using Bioptigen SD-OCT scanners, covering a 6.7 mm × 6.7 mm area centered on the fovea, producing volumetric scans of $1,000 \times 512 \times 100$ dimensions. Additionally, we also utilize an external validation dataset *(external)*, a data set of 458 B-scans from 159 unique participants from the Rotterdam Study (Hofman et al., 2007). This set was acquired using a Topcon system, obtaining 512 × 885 pixels or 512 × 650 pixels and covering a 6x6mm area centered on the macula.

**Training and Inference.** The *internal* dataset was divided into distinct sets for training (179 AMD, 71 normal), validation (10 AMD, 10 normal), and testing (80 AMD, 34

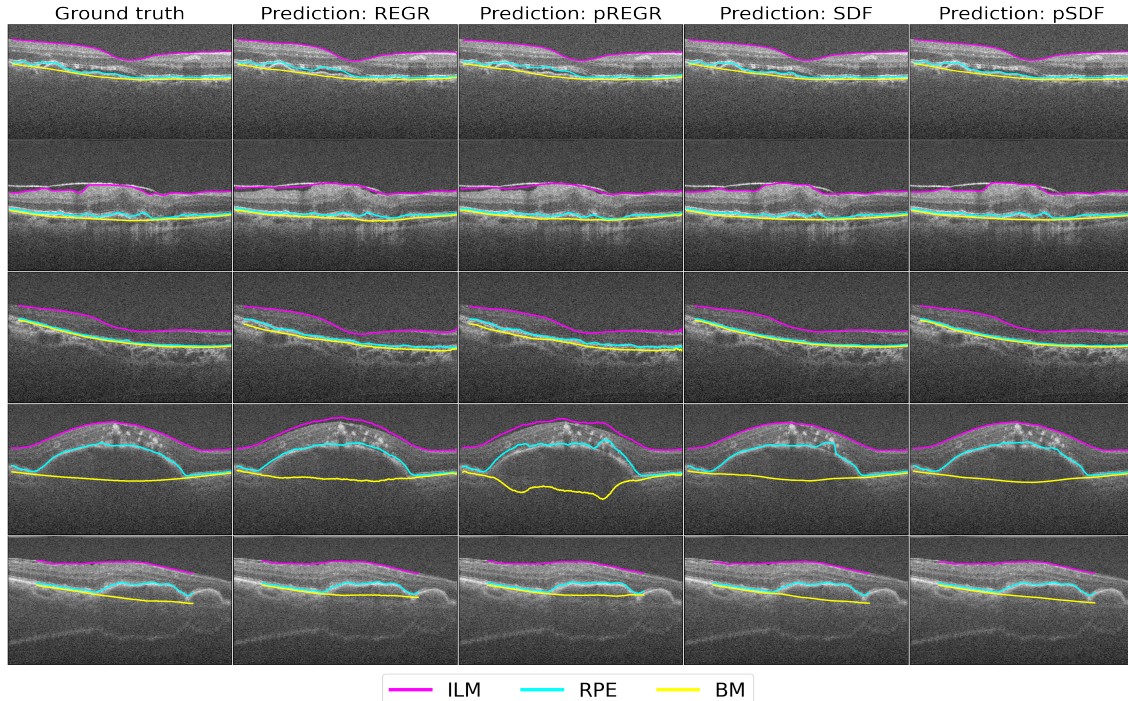

Figure 3: Segmentation examples for *REGR, pREGR, SDF, and pSDF*. The first and second row contains structural deformation due to AMD, and the third row presents a case of closely packed layers. Finally, last two rows show instances of Pigment Epithelial Detachment

normal). A comprehensive suite of randomized data augmentations was applied, including shearing, rotation, zooming, translation, intensity scaling, shifting, contrast adjustment, and the addition of Gaussian noise. For the *REGR* and *pREGR* methods, training uses the ground truth coordinates in the 2D image plane as provided by the dataset. The *SDF* and *pSDF* methods were trained using signed distance functions, which were constructed from these coordinates (see A.1). We use ResUnet++ (Jha et al., 2019) as the backbone architecture for all the SDF models (see A.3 & fig. 5 for details). The SDF models were trained with the negative log-likelihood loss with the addition of a clamping function (see A.4 for details). For robust extraction of the boundary from the SDFs, we employ a soft boundary extraction mechanism (see A.2) instead of directly solving for the zero-level.

**Experimental Validation.** To validate the effectiveness of our proposed approach, we evaluate the layer segmentation performance of *REGR/pREGR* and *SDF/pSDF* on both *internal* and *external* test sets using the Mean Absolute Error (MAE) in pixels. We compare our methods against previous studies (Sousa et al., 2021; Lou et al., 2020; Chen et al., 2019; Kugelman et al., 2018; Santos et al., 2019; Morelle et al., 2023), on the *internal* dataset. Additionally, in our framework, the estimation of uncertainty is geared towards gauging *aleatoric uncertainty*, focusing predominantly on the integrity and visibility of the segmented retinal layers. To validate this, we compute the correlation of the calculated uncertainty with layer integrity and visibility. We expect a higher value of uncertainty

in regions where the layer integrity is compromised or visibility is poor. To this end, we measure the average variance for each A-scan in randomly selected regions before and after being synthetically corrupted with noise such as shadow, blinking, speckle, and motion, naturally occurring in OCT scans due to various factors including disease progression and imaging noise. More about the noisy sample generation is detailed in A.5.

## 4. Result and Discussion

### 4.1. Layer Segmentation

The segmentation performance of different methods are detailed in Table 1. Our method, both in its probabilistic (*pSDF*) and non-probabilistic (*SDF*) forms, significantly boosted the performance over *REGR/pREGR* method, as evidenced by the improvement of the averaged MAE — approximately 2.38x for the *internal* and 2.4x for the *external* test set in the probabilistic version, and similar improvements in the non-probabilistic one. Moreover, the lower standard deviation suggests improved segmentation precision and consistency across B-scans. We also note that the introduction of uncertainty estimation leads to slightly better segmentation performance. This improvement can be attributed to how uncertainty prediction modulates the residual loss in Eq. 5. Effectively functioning as a data-adaptive regression modifier, it allows the network to adjust the impact of residuals (Kendall and Gal, 2017a). This is particularly beneficial for reducing the influence of erroneous labels, a common challenge in segmentation tasks, and, consequently, making the model more robust to noisy data.

Fig. 10 of the appendix shows a comparison of our method with previous studies, surpassing their reported performance. It is important to note, however, that these studies employed different training and test data partitions, and direct comparisons are not possible. Additionally, we present a comparative analysis featuring a pixel-wise segmentation method, as detailed in appendix B.1, alongside our approach.

Finally, Fig. 3 presents a series of five distinct examples. It is evident that the *SDF/pSDF* approach enhances segmentation precision, particularly in the presence of AMD-related abnormalities. As shown in the first two rows of Fig. 3, intricate structural deformation (such as drusen) were more accurately captured by *SDF/pSDF*. The third row demonstrates our method's proficiency in handling cases where the retinal layers are closely packed, showcasing enhanced delineation in such challenging scenarios. Furthermore, the last rows present cases of Retinal Pigment Epithelial Detachment (PED), a pathological condition that appears as domed elevations of the RPE layer (Todorich et al., 2012), posing significant segmentation challenges. Our approach handles such cases well compared to *REGR/pREGR*, demonstrating its robustness and improved accuracy.

### 4.2. Uncertainty Estimation

The quantitative results of our experiment with artificially distorted B-scans are shown in Fig. 4(a). These findings highlight that our approach consistently registers high un-

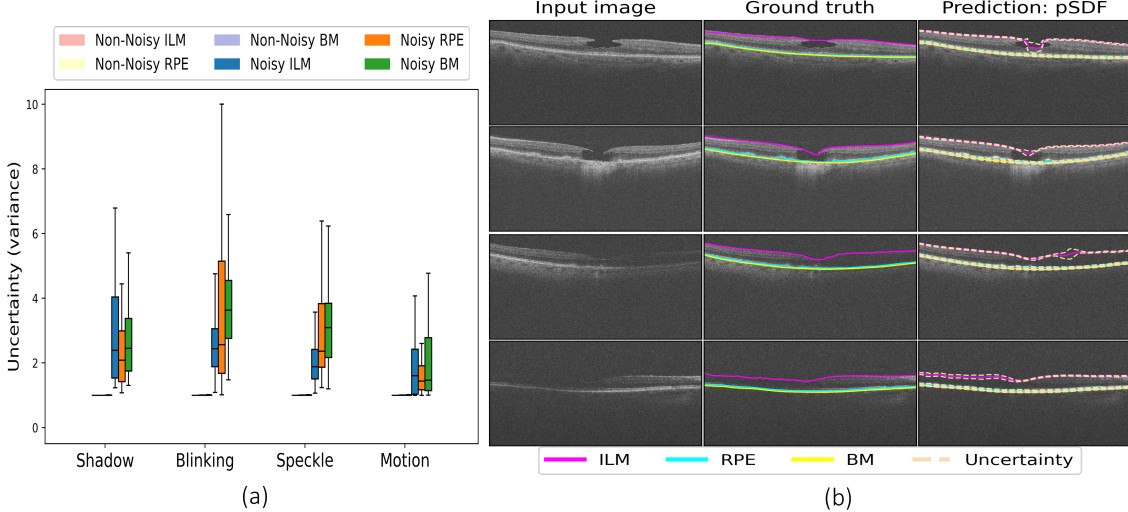

Figure 4: (a) Avg. per A-scan variance with *pSDF* for different types of noise and their non-noisy counterparts, (b) Uncertainty on pathological and noisy conditions with *pSDF*. The dotted line shows a +1 or -1 standard deviation.

certainty in the presence of all types of noise. Illustrative examples of such distortion are provided in the Appendix (Fig. 11). Additionally, a series of examples in Fig. 4(b) shows naturally occurring phenomena (The first and second rows contain instances of macular holes, third and fourth rows show partial layer invisibility.) that affect layer integrity and visibility. As shown, our model accurately produces larger uncertainty for disrupted or ambiguous regions. We also show a comparison with other popular uncertainty estimation approaches (namely Monte Carlo Dropout & Deep Ensemble), highlighting the superiority of our approach (see B.2). These results suggest that the quantified uncertainty in our model transcends mere segmentation confidence, potentially flagging pathological alterations tied to layer integrity. It can be particularly valuable for diseases affecting structural integrity, such as retinitis pigmentosa (RP) and geographic atrophy (GA), which especially cause disruption of the ellipsoid zone and thinning or attenuation of the retinal pigment epithelium, respectively. This uncertainty measurement can effectively highlight areas where the retinal structure is compromised, offering deep insights into disease progression.

## 5. Conclusion

In this paper, we concentrate on advancing retinal layer segmentation in OCT B-scans by focusing on two main aspects: improved segmentation of thin layers and enhanced uncertainty quantification. We introduce an SDF-based approach to achieve better layer shape representation, addressing the shortcomings of previous approaches and leading to improved segmentation. This representation also inherently leads to a more accurate estimation of the uncertainty concerning the layer shape, offering a deeper understanding of layer integrity and visibility in pathological and noisy conditions.

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

## Appendix A. Additional Materials

### A.1. Constructing Target Space

The ground truth distance function, which serves as the target for the neural network (NN) training, is generated from the segmented layers within 2D B-scans provided with the *internal* dataset. These layers, delineated by lines that denote boundaries, are transformed into distance functions. Since calculating the signed distance function (SDF) given a segmentation mask is non-trivial (Zhao, 2005), we use Danielsson's algorithm (Danielsson, 1980) to accomplish it. Note that by using Danielsson's algorithm for converting binary masks into signed distance maps, we ensure adherence to the Eikonal constraint, $\forall_{x,y} : \|\frac{\partial}{\partial y}d(x,y)\| = 1$, where $d$ represents the distance function. This algorithm computes the Euclidean distance from each pixel to the nearest object boundary, satisfying the Eikonal equation by maintaining a gradient magnitude of one in proximity to boundaries. Hence, our model implicitly learns to uphold the constraint, as the training data generation process inherently respects this fundamental geometric condition. The process is defined as follows:

In the domain $\Omega \subset \mathbb{R}^2$, the SDF $d^* : \mathbb{R}^2 \to \mathbb{R}$ assigns each pixel $p \in \mathbb{R}^2$ the shortest distance to the boundary $\partial\Omega$. For a binary mask $M$ indicating segmented layers by line pixels $\partial\Omega \subset M$, the function $d^*$ assigns each $p \in M$ to its nearest point in $\partial\Omega$. Formally, this is given by $d^*(p) = \min_{q \in \partial\Omega} \|p - q\|$, where $\| \cdot \|$ denotes the Euclidean norm. As the structures delineated by $\partial\Omega$ do not enclose any area, i.e., there are no inherent 'interior' or 'exterior' regions, leading to an absence of signed distances in the initial computation; all distances are non-negative. To incorporate the notion of sign, we introduce an orientation to the line $\partial\Omega$ by defining a directional vector $\overrightarrow{\boldsymbol{v}}$. This vector establishes a 'positive' direction above the line and a 'negative' direction below the line. The signed distance function $d^{gt}$ is then constructed by assigning a sign to the distances in $d^*$ based on the position of each pixel relative to $\partial\Omega$. This is expressed as:

$$d^{gt}(p) = \begin{cases} d^*(p) & \text{if } (p - q) \cdot \boldsymbol{v} \geq 0 \text{ for the closest } q \in \Omega, \\ -d^*(p) & \text{otherwise.} \end{cases} \tag{7}$$

In this way, $d^{gt}(p)$ represents the SDF, with the zero level set corresponding precisely to the line $\partial\Omega$, pixels above the line having positive values, and those below have negative values. The NN is thus trained to approximate this SDF, learning the spatial relations encoded within the signed distance function.

### A.2. Soft Boundary Extraction

Now having the SDF $d$ predicted by the network, we retrieve the retinal layer boundary coordinate by first applying a Gaussian mask to generate a weight mask $W$ for the whole OCT B-scan. The mask is calculated using the formula:

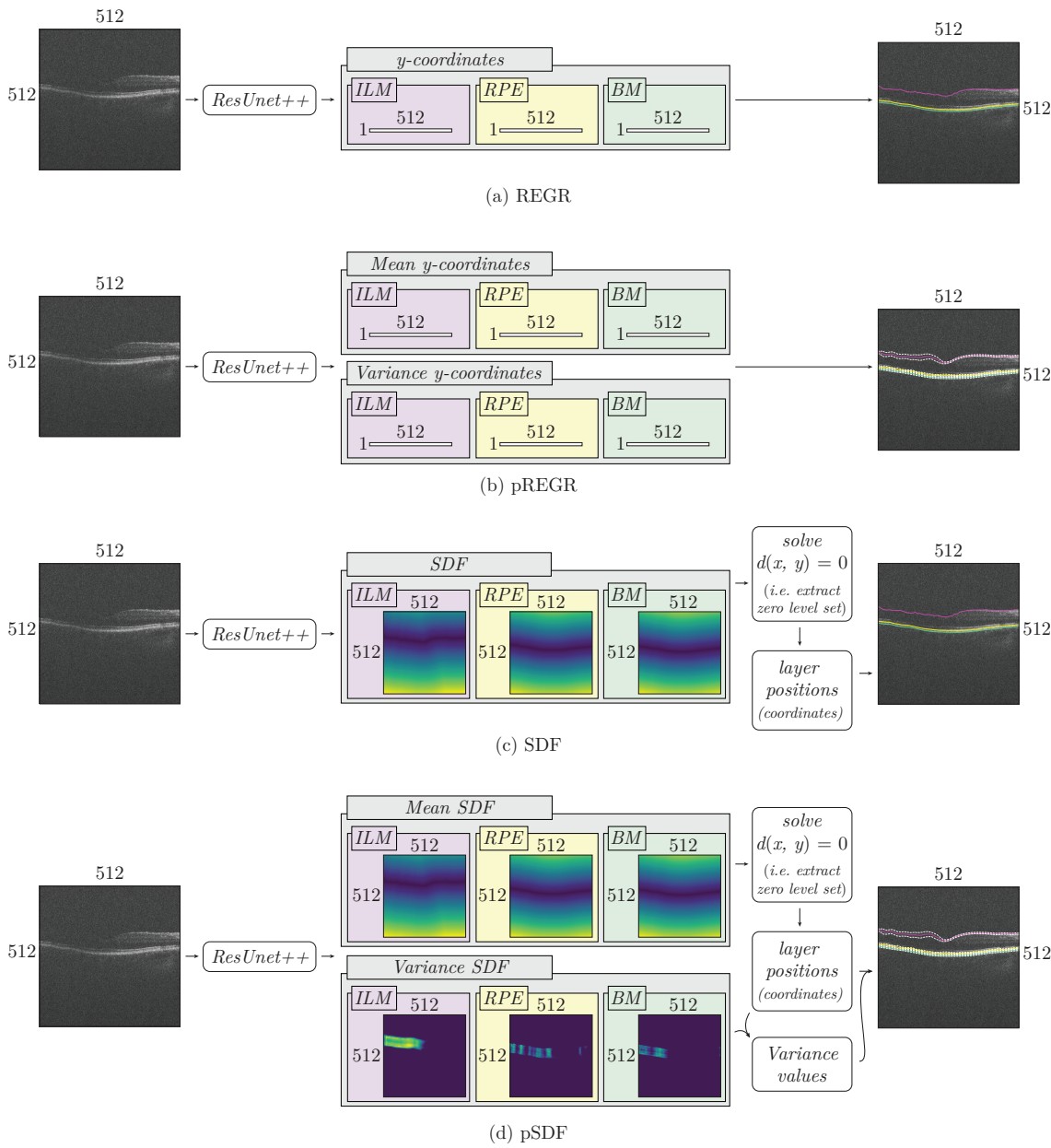

Figure 5: Schamatics for regression-based and our approach.

$$W_{xy} = \exp\left(-\frac{1}{2}\left(\frac{d(x,y) - level}{s_{const}}\right)^2\right), \tag{8}$$

where *level* is the SDF level that defines the boundary (typically zero) and $s_{const}$ is a predefined value. This masking operation serves to highlight the regions in the B-scan where the retinal layer boundary is likely to be located. Then the boundary is extracted by applying a weighted averaging scheme to the pixel indices, using $W$. For each column $x$ (corresponding to each A-scan), the boundary coordinate $y_x^{sdf}$ is computed as:

$$y_x^{sdf} = \frac{\sum_y y \cdot W_{xy}}{\sum_y W_{xy}}, \tag{9}$$

where $y$ runs over all pixel indices in the column. This essentially, transforms the implicit SDF representation into explicit coordinates, pinpointing the retinal layer boundaries.

### A.3. Deep Learning Model

For the *REGR* approach, we modify the ResUnet++ architecture to regress the retinal layer boundary coordinates, producing an output of $3 \times 512$ for $512 \times 512$ input B-scans as shown in fig. 5(a). This is primarily done by a non-expanding decoder for one of the spatial dimensions as explained by (Liefers et al., 2019). We use a significantly smaller model of 70 million parameters, in contrast to the 187 million parameters in (Liefers et al., 2019) yet experiments with their proposed architecture yield comparable results. In the *SDF* approach we adhere closely to the original ResUnet++ configuration except, for outputting three signed distance functions that mirror the input's spatial dimensions. For the uncertainty-aware variants, *pREGR* and *pSDF*, a new prediction head is integrated to predict the mean and variance for each column and pixel location, as shown in 5(b) & 5(d) respectively. Despite the architectural variations in *REGR/pREGR* and *SDF/pSDF*, we try to keep the models as close as possible to facilitate a fair comparison between methods.

### A.4. Loss Function

In alignment with our methodology, we optimize the Negative Likelihood (NLL) across all models, both probabilistic and deterministic. For the *REGR* approach, the NLL manifests simply as mean squared error (MSE) under the assumption of a Gaussian distribution with fixed variance. For *SDF* however, we slightly deviate since empirical observations indicate that an L1 loss results in superior segmentation fidelity compared to an L2 loss. Additionally, to increase the concentration capacity of the network on details near the surface (Park et al., 2019), we introduce a 'clamp' function, restricting the error within a specified range:

$$\mathcal{L}_{\text{SDF}}(\theta) = \sum_{n,x,y} |\text{clamp}(d^{gt}(x,y),\delta) - \text{clamp}(f^{sdf}[I_n;\theta](x,y),\delta)|, \tag{10}$$

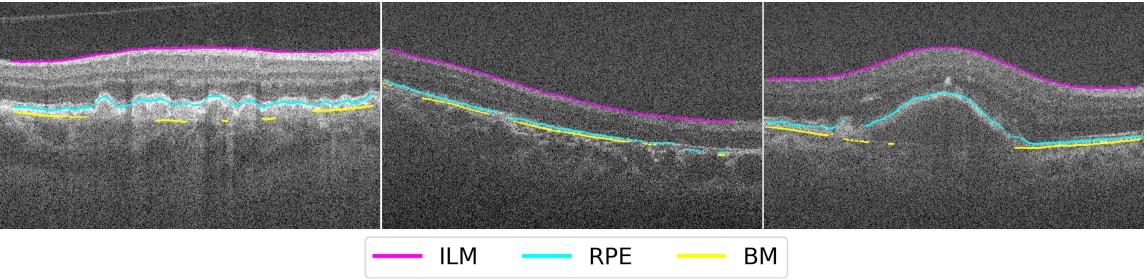

Figure 6: Examples of prediction with the pixel-wise segmentation approach.

The function $clamp(x, \delta) := \min(\delta, \max(-\delta, x))$ introduces the parameter $\delta$ to control the distance from the surface over which we expect to maintain a metric SDF.

In the probabilistic counterpart, we optimized as it's shown in 5. Again for *pSDF*, we adapt to its clamped version as following:

$$\mathcal{L}_{\mathrm{pSDF}}(\theta) = \sum_{n,x,y} -\frac{1}{2}\left[\frac{(\mathrm{clamp}(d^{gt}(x,y),\delta) - \mathrm{clamp}(f^{sdf}_{\mu}[I_n;\theta](x,y),\delta))^2}{f^{sdf}_{\sigma}[I_n;\theta]^2(x,y)} + \ln f^{sdf}_{\sigma}[I_n;\theta]^2(x,y)\right], \quad (11)$$

Here the chosen $\delta$ values are typically between 28-30. We empirically find a larger clamp value results in less precise segmentation bounderies whereas a smaller value leads to a generalization problem resulting in overall reduced performance, especially on the *external* dataset.

### A.5. Noisy Sample Generation

Here we explain in detail how we generate the noisy sample with different types of noise and what some analogous naturally occurring phenomena in OCT to these noises.

**Shadow artifact.** Shadowing can naturally occur due to blockages or opacities in ocular media, often caused by cataracts or hemorrhages, leading to darkened or obscured regions in the scans (Duker et al., 2021).

To add the shadow artifact within a specific B-scan, we adhere to the procedure outlined by (de Vente et al., 2023). For any given B-scan, each constituent A-scan $a_x$ undergoes an individualized modification process. Here, $x$ spans the totality of A-scans, collectively numbered as $\mathcal{X}$. The transformation of each A-scan $a_x$ is governed by the shadow function $S$, defined as $S(a_x) = a_x \cdot (1 - s(x))$, where the term $s(x)$ is derived from a Gaussian probability density function: $s(x) = \frac{1}{\sigma\sqrt{2\pi}}e^{-\frac{1}{2}\left(\frac{x-\mu}{\sigma}\right)^2}$. Parameters $\mu$ and $\sigma$ are randomly determined within the bounds $[0, \mathcal{X}]$ and $\left[\frac{\mathcal{X}}{4}, \frac{3\mathcal{X}}{4}\right]$ respectively.

**Blinking artifact:** Blinking artifacts are analogous to motion artifacts caused by patient movement or blinking during the scan, resulting in discontinuities or stripes in the B-scan(Chhablani et al., 2014).

To introduce artificial blinking artifacts into B-scans, we commence with a B-scan where all pixel intensities are set to zero. Subsequently, we superimpose additive Gaussian noise onto this B-scan. The mean of the Gaussian distribution was set to the median pixel value

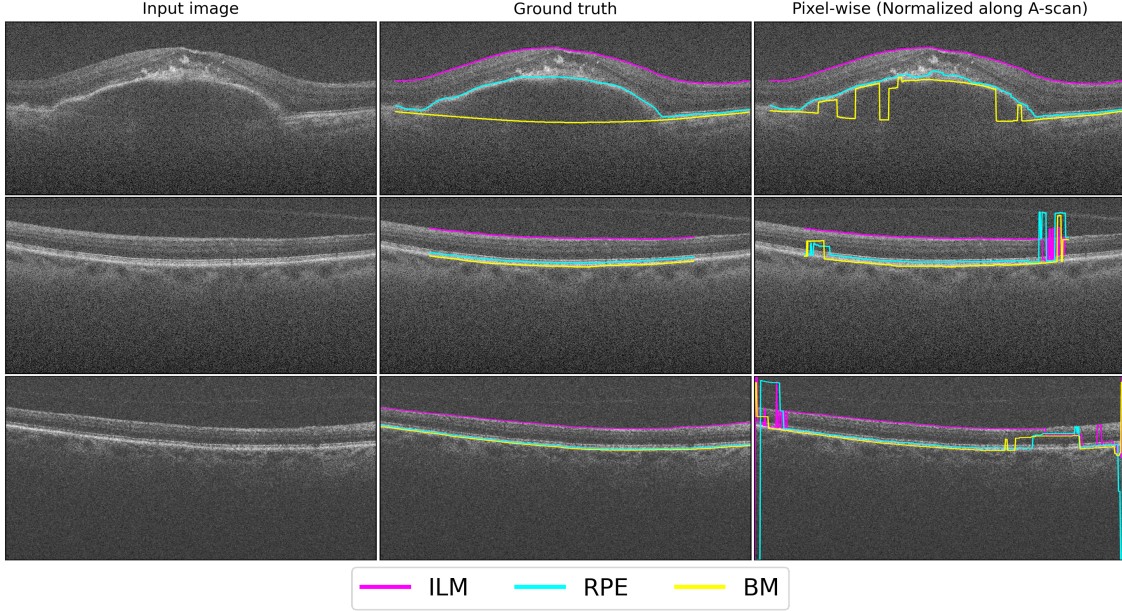

Figure 7: Examples of prediction with the pixel-wise segmentation approach. Predicted probabilities are normalized along each A-scan.

observed across the entire OCT scan, while the standard deviation was matched to the standard deviation inherent to the OCT scan itself.

**Speckle artifact:** Speckle is an inherent property of OCT imaging, arising from the coherent nature of the light source and the interference of backscattered light, manifesting as granular noise that can mask underlying retinal structures(Curatolo et al., 2013).

To replicate the characteristic granularity of speckle noise in B-scans, a binary modification of pixel intensities is employed. A probability $p$, uniformly selected from the interval [0.4, 0.6], determines the binary state of each pixel. Accordingly, pixels are stochastically assigned either the maximum or minimum intensity values present in the B-scan, thereby producing the sharp, speckled contrasts characteristic of speckle noise.

**Motion artifact:** Similar to blinking, motion noise in OCT can be attributed to micro-movements of the patient or the eye during scanning, leading to blurring or smearing of the B-scan (Chhablani et al., 2014).

To simulate motion artifacts in B-scans, we implement a localized pixel displacement. For each horizontal cross-section of the B-scan, a random shift is applied, with the shift magnitude uniformly drawn from the interval $[-\Delta_{max}, \Delta_{max}]$, where $\Delta_{max}$ denotes the maximum allowed pixel shift. This shifting process introduces a realistic representation of motion blur, akin to the motion artifacts.

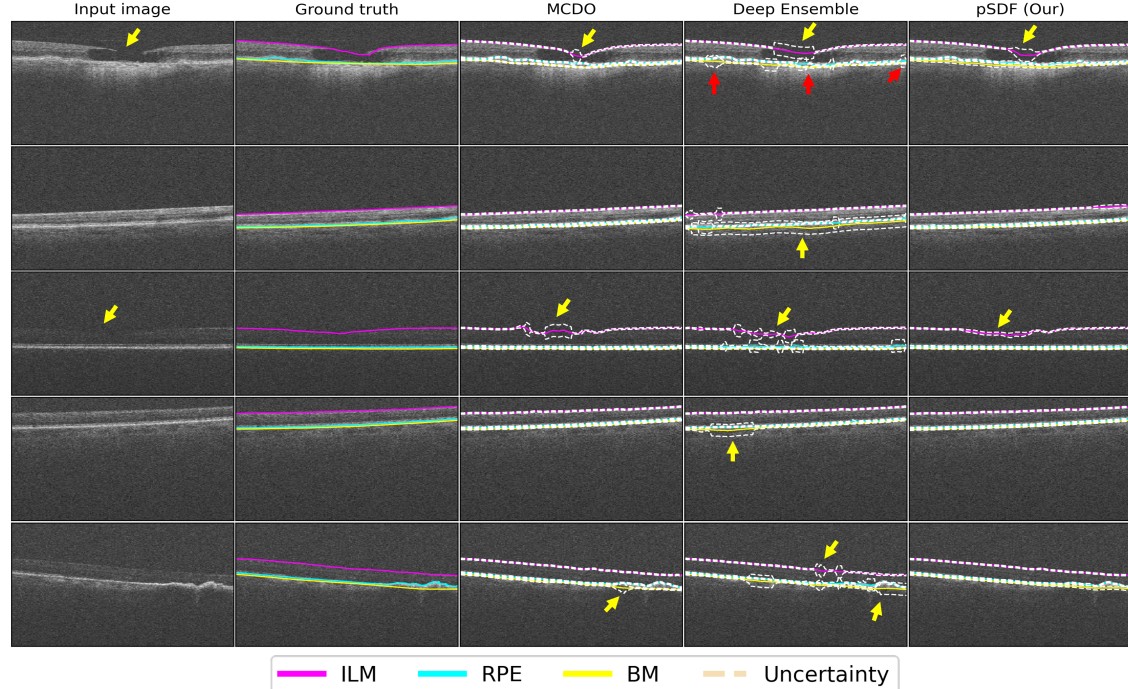

Figure 8: Uncertainty estimation examples for MCDO, Deep ensemble, and pSDF (our). The dotted line shows +1 or -1 standard deviation. In the first row, we expect to see high uncertainty only in the region where the macular hole (layer not present) is. However, MCDO shows high uncertainty in a region that is substantially smaller than the size of the macular hole, while the Deep Ensemble method exhibits overestimation (highlighted with yellow arrows). Conversely, the region with high uncertainty from our approach is accurately co-located with the macular hole. Furthermore, the Deep Ensemble introduces unwanted high uncertainty areas within the BM and RPE layers—marked by a red arrow—despite clear imaging features. This issue recurs in the second and fourth rows (yellow arrows). In the third row, a segment of the ILM near the yellow marker is obscured due to inferior image quality, where a consistently high uncertainty was anticipated. Instead, both Deep Ensemble and MCDO exhibit alternating high and low uncertainty patterns. Similarly, the fifth row evidences unexpected high uncertainty regions produced by both MCDO and Deep ensemble (yellow arrows).

## Appendix B. Additional Results

### B.1. Pixel-wise Segmentation

A pixel-wise model has also been trained to evaluate performance against our proposed approach. We use the same architecture described for *SDF* approach and cross-entropy loss to train the model. However, this immediately leads to the problem shown in Fig. 6, as some of the pixels that belong to the layer boundaries are classified as background pixels. The problem highlights the fact that due to the thin layer structure, using a pixel-wise cross-entropy loss fails to provide a robust supervisory signal to train the model properly as discussed in the section 1. The problem of disconnected lines can partially be alleviated

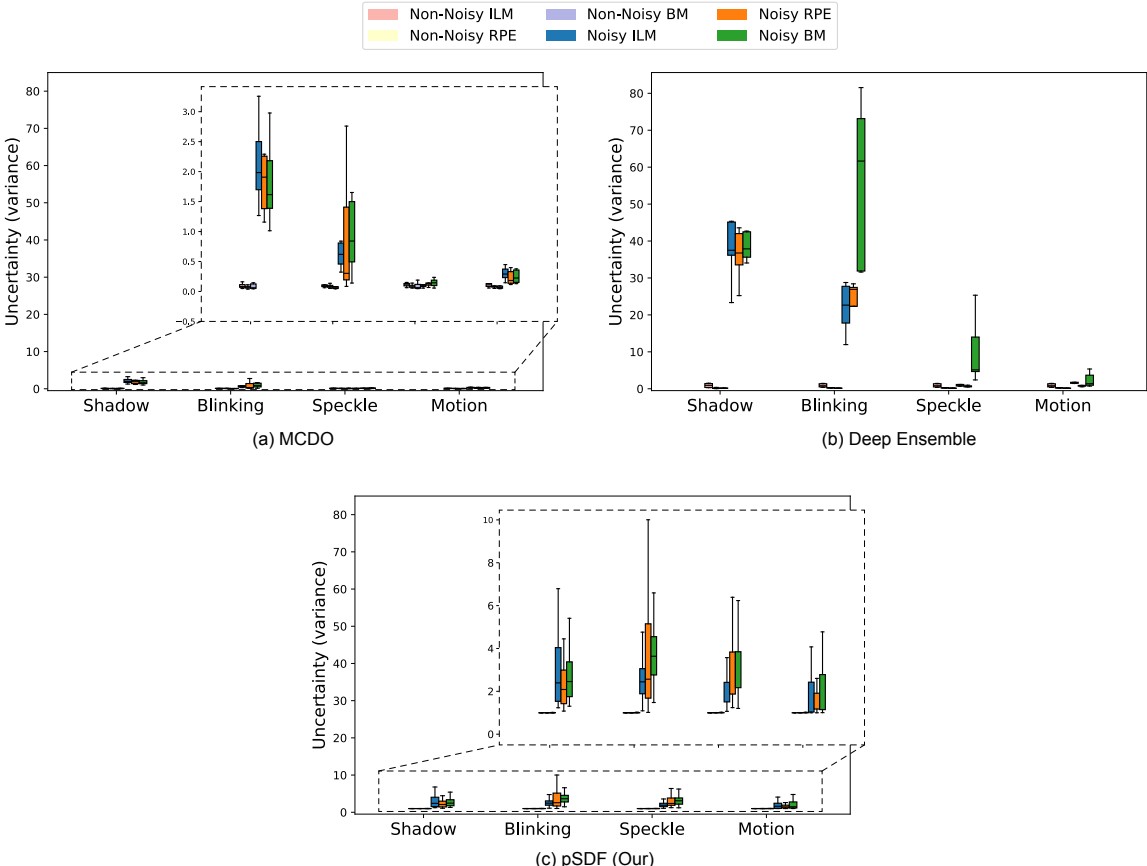

Figure 9: Avg. per A-scan variance for different types of noise and their non-noisy counterparts, shown for the three different uncertainty estimation techniques (a) MCDO, (b) Deep Ensemble, and (c) pSDF.

by simply applying softmax to each A-scan (per column of the 2D image) and taking the coordinate with the highest probability. This ensures a single coordinate for each layer per A-scan. This allows us to compute the mean absolute error to compare with other approaches. However, this approach is also not stable in terms of prediction as it shows oscillating behavior shown in Fig. 7. Tab. 2 outlines the performance of this approach, which performed nearly ≈7x and ≈10x worse than our proposed method on *internal* and *external* datasets respectively.

## B.2. Uncertainty Comparison

We compare our uncertainty estimation with more widely used approaches such as Monte Carlo Dropout (MCDO) (Kendall and Gal, 2017b) and Deep Ensemble (Lakshminarayanan et al., 2017). We again use the same experimental setup as our proposed approach. Addi-

tionally, to estimate the performance we consider 5 ensemble members trained from scratch for deep ensemble, and we sample 15 times for each input image for MCDO. We make the following observations when we compare to our method.

**Segmentation Performance.** The segmentation performance of the Deep Ensemble aligns closely with that of our method, as detailed in Tab. 2. Conversely, the Monte Carlo Dropout (MCDO) approach experienced a minor decline in performance across both internal and external datasets.

Table 2: Layer segmentation performance for additional methods (measured with mean absolute error (MAE) in pixels).

| Method | Layer | MAE (internal) | (external) |
|---|---|---|---|
| Pixel-wise | ILM | $1.47_{\pm 2.93}$ | $8.55_{\pm 6.47}$ |
| | RPE | $2.51_{\pm 3.77}$ | $16.39_{\pm 11.09}$ |
| | BM | $6.86_{\pm 10.54}$ | $10.01_{\pm 6.57}$ |
| | Avg. | 3.62 | 11.65 |
| Deep ensemble | ILM | $0.32_{\pm 0.43}$ | $1.10_{\pm 1.10}$ |
| | RPE | $0.58_{\pm 0.36}$ | $1.60_{\pm 1.46}$ |
| | BM | $0.77_{\pm 0.36}$ | $1.39_{\pm 0.96}$ |
| | Avg. | 0.56 | 1.37 |
| MCDO | ILM | $0.35_{\pm 0.40}$ | $1.25_{\pm 0.70}$ |
| | RPE | $0.63_{\pm 0.41}$ | $1.52_{\pm 1.33}$ |
| | BM | $0.79_{\pm 0.32}$ | $1.36_{\pm 0.77}$ |
| | Avg. | 0.59 | 1.38 |
| pSDF (Our) | ILM | $0.32_{\pm 0.44}$ | $1.63_{\pm 0.78}$ |
| | RPE | $0.59_{\pm 0.40}$ | $1.02_{\pm 1.47}$ |
| | BM | $0.75_{\pm 0.29}$ | $1.27_{\pm 0.82}$ |
| | Avg. | **0.55** | **1.30** |

**Better Uncertainty.** To compare the uncertainty with our approach, we measure the average variance for each A-scan in randomly selected regions before and after being synthetically corrupted with noise patterns as explained in section 3. The results can be found in Fig. 9 (a-b). We note here that both MCDO and Deep ensemble remain very sensitive to shadow and blinking noise but insensitive to other types of noise, leading to overall unbalanced uncertainty estimation. In contrast, our method produces more balanced uncertainty across the board as shown in Fig. 9 (c). Further qualitative evaluation also reveals that both MCDO and Deep ensemble often produce erroneous uncertainty estimates (Fig. 8). For example, in cases where the layers are well visible, high uncertainties can be produced by these approaches. Furthermore, the extent of regions in which higher uncertainty is expected due to certain pathologies or imaging artifacts is often underestimated or overestimated.

**Computational Cost** In our approach, we deploy an ensemble of five members, each requiring independent training. This necessitates executing five forward passes during in-

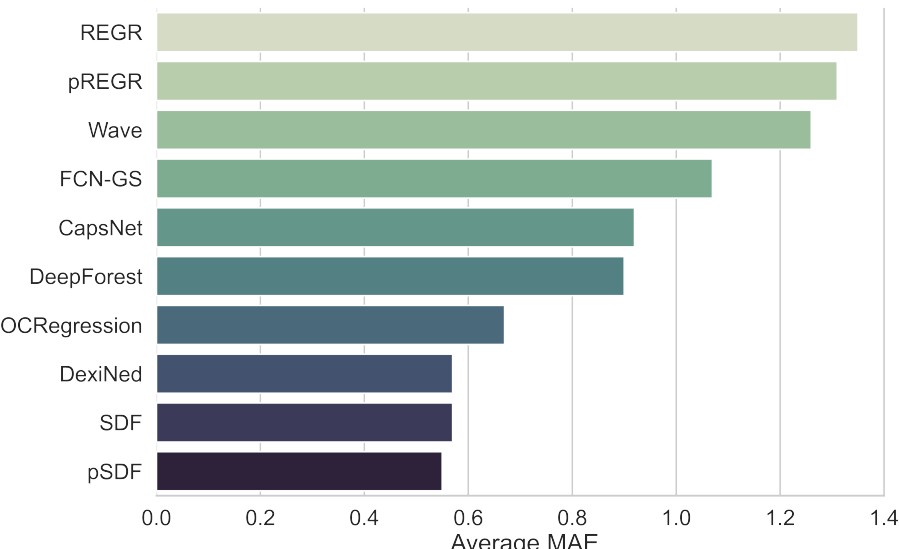

Figure 10: Layer segmentation performance comparison with other studies, measured with mean absolute error (MAE) in pixels. FCN-GS (Kugelman et al., 2018), CapsNet (Santos et al., 2019), DeepForest (Chen et al., 2019), Wave (Lou et al., 2020), DexiNed (Sousa et al., 2021), REGR (Liefers et al., 2019), OCRegression (Morelle et al., 2023)

ference, quintupling the computational demand for both the training and inference phases. Additionally, the requirement to store the weights of five distinct models increases the memory footprint. Also, the MCDO technique predominantly inflates the computational cost during inference, which in our implementation, increases fifteenfold as we perform fifteen sampling operations for each input image.

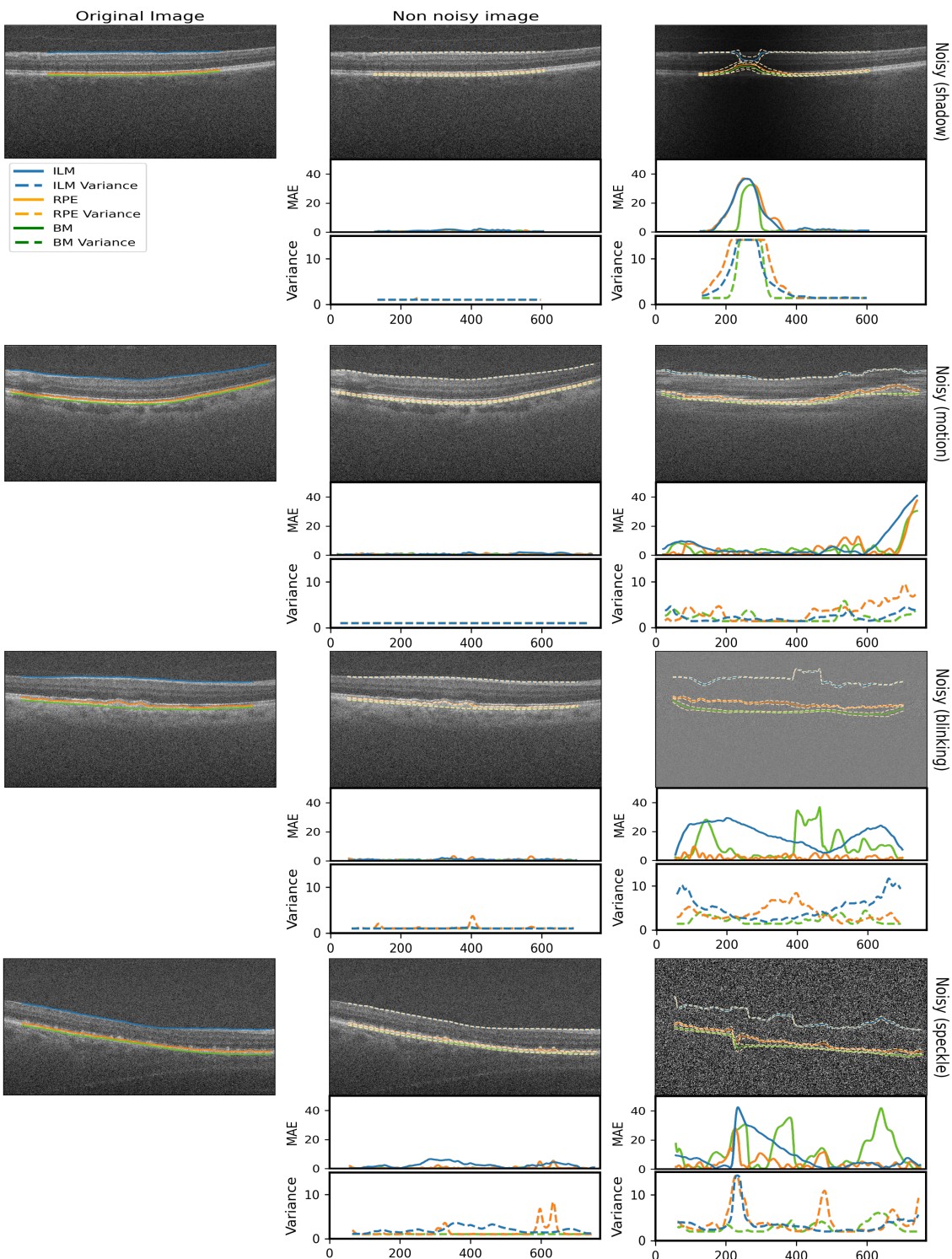

Figure 11: Prediction with uncertainty for artificially distorted samples.(MAE=Mean absolute error)

