# OpenReview forum: "Uncertainty-aware retinal layer segmentation in OCT through probabilistic signed distance functions"
_MIDL.io/2024/Conference — MIDL 2024 Poster_

### Official Review · Reviewer_nGsx · 2024-02-22

**Confidence:** 5
**Preliminary Rating:** 4
**Recommendation:** Poster
**Final Rating:** 4

**Summary:**

This paper refines the segmentation by predicting a SDF that effectively parameterizes the retinal layer shape via level set. They extended Gaussian modeling for aleatoric uncertainty estimation in their framework. In general, the motivation is sufficient, and the methods are clearly defined, but the comparison with existing methods and literature review is insufficient.

**Strengths:**

They present a redefined approach to retinal layer boundary segmentation with SDF.
They  extend Gaussian modeling for aleatoric uncertainty estimation in their framework.
They will release the code publicly.

**Weaknesses:**

1. The comparative methods are all existing regression methods. Have other uncertainty estimation methods [1-3] and alternative foundational frameworks [4-5] not been considered?

2. The authors claim to have addressed the challenges of thin layer segmentation and the lack of geometric grounding. However, there is no presentation or demonstration of these two challenges.

[1] MCdropout: What uncertainties do we need in bayesian deep learning for computer vision?
[2] ensemble: Simple and scalable predictive uncertainty Estimation using deep ensembles.
[3] Evidential-based method: Reliable joint segmentation of retinal edema lesions in oct images
[4] nnUnet: a self-configuring method for deep learning-based biomedical image segmentation
[5] Transformer-based: An image is worth 16x16 words: Transformers for image recognition at scale

**Detailed Comments:**

1. This draft uses regression for segmentation, as indicated in Equation 6. Has the consideration of common segmentation loss functions, such as Dice loss, been taken into account as supplementary options?
2. The comparative methods are all existing regression methods. Have other uncertainty estimation methods [1-3] and the substitution of different foundational frameworks [4-5] been considered?
3. The authors claim to have addressed the challenges of thin layer segmentation and the lack of geometric grounding. Can these two challenges be illustrated with example images?
[1] MCdropout: What uncertainties do we need in bayesian deep learning for computer vision?
[2] ensemble: Simple and scalable predictive uncertainty Estimation using deep ensembles.
[3] Evidential-based method: Reliable joint segmentation of retinal edema lesions in oct images
[4] nnUnet: a self-configuring method for deep learning-based biomedical image segmentation
[5] Transformer-based: An image is worth 16x16 words: Transformers for image recognition at scale

**Justification Of Final Rating:**

Thanks for the author's reply, benefited a lot.
Based on the author's reply, I have no updated comments and maintain the original score.
Based on the author's reply, I have no updated comments and maintain the original score.

**Justification Of The Preliminary Rating:**

The basis of the preliminary rating mainly comes from the following aspects: the review of uncertainty estimation methods, uncertainty estimation indicators, the application and experiment of uncertainty estimation methods.

**Questions To Address In The Rebuttal:**

1. This draft uses regression for segmentation, as indicated in Equation 6. Has the consideration of common segmentation loss functions, such as Dice loss, been taken into account as supplementary options?
2. The comparative methods are all existing regression methods. Have other uncertainty estimation methods [1-3] and the substitution of different foundational frameworks [4-5] been considered?
3. The authors claim to have addressed the challenges of thin layer segmentation and the lack of geometric grounding. Can these two challenges be illustrated with example images?
[1] MCdropout: What uncertainties do we need in bayesian deep learning for computer vision?
[2] ensemble: Simple and scalable predictive uncertainty Estimation using deep ensembles.
[3] Evidential-based method: Reliable joint segmentation of retinal edema lesions in oct images
[4] nnUnet: a self-configuring method for deep learning-based biomedical image segmentation
[5] Transformer-based: An image is worth 16x16 words: Transformers for image recognition at scale

**Special Issue:**

No

---

> ### Author Response · Authors · 2024-03-16
> **Response to reviewer (nGsx)**
>
> We would like to thank the reviewer (nGsx) for providing detailed feedback. We have implemented several modifications in response. We are confident that these changes have improved the clarity and coherence of our work. In the following sections, we address each of the points raised by the reviewer,
>
>
> - **"Has the consideration of common segmentation loss functions, such as Dice loss, been taken into account as supplementary options?"**
>
>   We have experimented with both cross-entropy loss and Dice loss. In our experiment, we found that training was unstable with the Dice loss, but we were able to train a model with a cross-entropy loss. However, this approach leads us to the problem of disconnected layer boundaries (please see newly added Fig.6), as some of the pixels that belong to the layer boundaries are classified as background pixels. The problem of disconnected lines can be alleviated by applying softmax to each A-scan (each column of the 2D image) and taking the coordinate with the highest probability. This ensures a single coordinate for each layer per A-scan. But this is also unstable as it produces oscillation in the predicted line (please see newly added Fig.7). As for the performance compared to our approach pixel-wise approach performs approximately 7x and 10x worse on *internal* and *external* datasets.
>
>   In response to your feedback, we have added section B.1 in the Appendix detailing the pixel-wise segmentation approach.
>
> - **"Have other uncertainty estimation methods been considered?"**
>
>   In response to your feedback, we have included experimentation (B.2 of the Appendix) on how the Monte Carlo Dropout and Deep ensemble stack against our method. We note the following findings: (1) Segmentation performance of the Deep ensemble is on par with our method (please see Tab.02) whereas MCDO showed a slight performance drop, (2) Both MCDO and Deep ensemble remain very sensitive to certain types of noise (e.g. shadow, blinking) at the same time insensitive to other types of noise (please see Fig.9). Leading to overall unreliable uncertainty estimation (3) Carefully qualitative evaluation also reveals that both MCDO and deep ensemble often produces erroneous uncertainty (please see Fig.8) in challenging cases even though the layer features are detectable.
>
>   In conclusion, we found neither approach brings advantages in terms of segmentation and uncertainty whereas both Deep ensemble and MCDO result in 5x-15x (either inference or both training and inference) increases in computational requirements.
>
> - **"The comparative methods are all existing regression methods. Have other substitutions of different foundational frameworks been considered?"**
>
>   As mentioned in one of the previous issues, we have added a comparison with a pixel-wise segmentation trained with cross-entropy loss to provide more context to our approach and found it to be performed approximately 7x and 10x worse on *internal* and *external* datasets (please see tab.2).
>
>   As for other types of architecture, this has not been considered since the principles presented in the paper are agnostic to the particular type of architecture used and should easily extend to other variants of networks.
>
> - **"The authors claim to have addressed the challenges of thin-layer segmentation and the lack of geometric grounding. Can these two challenges be illustrated with example images?"**
>
>   The challenge of thin-layer segmentation in the context of the traditional approach (pixel-wise with dice/cross-entropy) can be seen in Fig.6 (newly added), as it highlights the difficulty of training a model in the presence of small/thin structure due to sparsity of robust supervisory signal. A possible solution is to use a regression model to predict a coordinate for each A-scan (columns in 2D image), i.e., we obtain a 1D curve from a 2D image. This has the disadvantage that somehow a conversion between different data structures (from 2D data to a 1D curve) needs to take place, and this comes at the cost of losing correspondence between predicted coordinates and the actual image coordinates. In other words, we lose the grounding of the output with respect to the input as now there is no way of telling how localized image features correspond to the coordinate predictions. It complicates relating inaccuracies/failures to specific input image features. This problem of a lack of correspondence is not faced with pixel-wise segmentation, however, as noted earlier, pixel-wise segmentation suffers from other issues.
>
>   Both of these challenges are addressed by our approach in the following way: (1) during training the distance GNLL in combination with SDF provides a dense and robust supervisory signal to train the model, and (2) we reacquire the one-to-one pixel correspondence in a fully convolutional architecture (UNet type), assigning an SDF value and uncertainty to each pixel.

---

### Official Review · Reviewer_2R9g · 2024-02-26

**Confidence:** 4
**Preliminary Rating:** 4
**Recommendation:** Oral
**Final Rating:** 4

**Summary:**

This paper proposes a method for segmenting the several layers of optical coherence tomography scans. The heart of the proposed method is a signed distance function with uncertainty. The method is trained and validated on two different publicly available datasets and the results are shown to outperform simpler methods.

**Strengths:**

This paper is well-written, the model is simple while still mathematically well-founded and the results look convincing. I generally like the paper, and have nothing more to add, nothing more to add....

**Weaknesses:**

It seems that a comparison with the proposed representation is missing with Dong et al., "PSDF Fusion: Probabilistic Signed Distance Function for On-the-fly 3D Data Fusion and Scene Reconstruction", ECCV 2018. In particular, this article's equation (3) with Dong's equation (1) seem very similar. I also find the notation of \Phi_\mu and \Phi_\sigma too subtle for easy reading. Also, the color of the curves plotted on top of the images in Figure 3 and similar is very difficult to see, which makes it difficult to appreciate the comparison.

**Detailed Comments:**

I have no further comments.

**Justification Of Final Rating:**

The response to my questions is sufficient. However, in the light of the uncertainty about the novelty, my final rating remains the same. I have nothing more to add....................................

**Justification Of The Preliminary Rating:**

I like the paper, but there seems to be relevant overlap with the literature wrt. novelty on the probabilistic signed distance function. I have no further comments. I have no further comments. I have no further comments.

**Questions To Address In The Rebuttal:**

Please compare with Dong et al., please update the notation for mean and sigma networks, please update Figure 3, etc.

**Special Issue:**

Yes

---

> ### Author Response · Authors · 2024-03-16
> **Response to reviewer (2R9g)**
>
> We would like to thank the reviewer (2R9g) for providing detailed feedback. We have implemented several modifications in response. We are confident that these changes have improved the clarity and coherence of our work. In the following sections, we address each of the points raised by the reviewer,
>
>
> - **"Please compare with Dong et al.,"**
>
>  The referenced work addresses the challenge of reconstructing a true Signed Distance Function (SDF) from noisy measurements, considering two types of noise: inaccuracies in SDF measurements and the distinction between inliers (part of the shape) and outliers (noise). Their approach includes modeling the SDF measurement inaccuracies with a Normal distribution and employing a Beta distribution for outlier modeling leading to a mixture model. While we share the application of a probabilistic framework to SDF (utilizing a normal distribution for the mean SDF and its standard deviation), our contexts differ significantly. The author focuses on fitting an SDF to noisy observations of an SDF, whereas our aim is to deduce an SDF directly from image data. I.e., our method is an inference problem (deduces an SDF from image data), whereas their method is a reconstruction/denoising problem (filters out noise from an observed SDF). Consequently, a practical comparison is not possible, however, in light of the above discussion, we have added the study of Dong et al. to our related work section (which is embedded within the Introduction) and thank the reviewer for bringing it to our attention.
>
> - **"please update the notation for mean and sigma networks, please update Figure 3, etc."**
>
>  We have updated the notation for the neural networks from $\Phi$ to $f$ for ease of reading however we would still like to keep the sub and superscripts to maintain the consistency between the notation of the neural network across all the approaches (REGR,pREGR, SDF, and pSDF).
>
> We have also updated Fig.3 along with all the other figures for better visuals.

---

### Official Review · Reviewer_R8tK · 2024-02-26

**Confidence:** 3
**Preliminary Rating:** 3
**Final Rating:** 4

**Summary:**

This paper proposes using probabilistic SDF to perform retinal layer segmentation in OCT.
The main idea is to predict the mean and standard deviation of the pixelwise SDF using a ResUNet++.
Evaluations are mostly done against regression models that directly predict the a layer coordinate per vertical scan (A-scan), using two different datasets.

**Strengths:**

Overall, the paper reads well.
Evaluation methods make sense given the task and the proposed method. An "external" dataset (which presumably means a dataset from a different source that the model has never seen during training/validation) was used to perform a second round of evaluations.
Performance boost from the baseline is clear.

**Weaknesses:**

The model could have been easily compared against segmentation methods, but it is only compared against regression methods.
The idea of using SDF instead of segmentation is not entirely novel. Here is one example (but there should be many more): Xue, Yuan, et al. "Shape-aware organ segmentation by predicting signed distance maps." Proceedings of the AAAI Conference on Artificial Intelligence. Vol. 34. No. 07. 2020.
The uncertainty estimate  visualization in Fig. 4b is unclear. This makes it hard for us to acknowledge its usefulness.
The claim that "a neural network predicts a SDF with the property that it satisfies the Eikonal constraint" is unsubstantiated. Based on the implementation descriptions, it sounds like it was just a straightforward neural network prediction without any constraints on the predicted output. It seems speak more to the way that the SDF training data were generated.

**Detailed Comments:**

Figure 1 seems to be describing the idea of SDF and isosurface extraction. A more useful diagram may have been a schematic of how the probabilistic SDF model was implemented (2D image -> ResUnet++ -> pixelwise prediction of mean & std of SDF) vs. regression or other methods.

**Justification Of Final Rating:**

Thank you to the authors for carefully addressing the comments.
Although the main section hasn't changed significantly, many of the comments have been addressed through additional contents in the appendix. The quality of the original submission was also good enough for a higher rating given the clarifications.

**Justification Of The Preliminary Rating:**

The paper showcases the usefulness of using SDF to improve the OCT segmentation accuracy, and also shows that we can use the basic alleotoric uncertainty implementation to predict sensible uncertainties for corrupted inputs.
However, comparisons against segmentation methods, which would have been a more apples-to-apples comparison with the SDF approach, are missing. The idea of using SDF instead of segmentation labels is also not entirely new.
The presentation of the material could have been improved with more illustrative figures describing the baselines and the proposed method, as well as clearer visualization of the uncertainties.

**Questions To Address In The Rebuttal:**

Was the SDF model compared against segmentation models? How do the results compare?
What is Fig. 4b trying to show? Specifically, what is the meaning of the dotted line?
How is the Eikonal constraint satisfied from the ResUnet++ prediction?

---

> ### Author Response · Authors · 2024-03-16
> **Response to reviewer (R8tK)**
>
> We would like to thank the reviewer (R8tK) for providing detailed feedback. We have implemented several modifications in response. We are confident that these changes have improved the clarity and coherence of our work. In the following sections, we address each of the points raised by the reviewer,
>
>
> - **"The idea of using SDF instead of segmentation is not entirely novel."**
>
>   We recognize the reviewer's observation regarding the application of Signed Distance Functions (SDF) in segmentation tasks, a point we have acknowledged by citing relevant prior work in the introduction of our paper. However, we wish to highlight two critical distinctions that underscore the novelty and contribution of our work: (1) utilization of the signed distance function (SDF) approach to tackle thin structure segmentation (in this case one-pixel width retinal layer boundary) has not been shown in the literature. (2) Although other works have introduced a probabilistic framework for SDF, their approach did not extend to the uncertainty in segmentation or shape inference challenges. Our work advances the application of probabilistic SDF to measure uncertainties in such a scenario (e.g., segmentation/shape inference).
>
>
>     These points now have been elucidated more clearly at the conclusion of the introduction section.
>
> - **"Was the SDF model compared against segmentation models? How do the results compare?"**
>
>   The SDF model was compared with a pixel-wise segmentation model trained with cross-entropy loss. However, this approach leads to the problem of disconnected layer boundaries (please see newly added Fig.6), as some of the pixels that belong to the layer boundaries are classified as background pixels. The problem of disconnected lines can be alleviated by applying softmax to each A-scan (each column of the 2D image) and taking the coordinate with the highest probability. This ensures a single coordinate for each layer per A-scan. But this is also unstable as it produces oscillation in the predicted line (please see newly added Fig.7). As for the performance compared to our approach pixel-wise approach performs approximately 7x and 10x worse on *internal* and *external* datasets.
>
>   In response to your feedback, we have added section B.1 in the Appendix detailing the pixel-wise segmentation approach.
>
> - **"What is Fig. 4b trying to show? Specifically, what is the meaning of the dotted line?"**
>
>   Fig. 4b in our paper aims to illustrate the uncertainty (+1 or -1 standard deviation) for each A-scan with the dotted line. i.e., the measure reflects how much the predicted coordinate can deviate from the predicted mean in the vertical direction.
>
>   However, we acknowledge that it is not explicitly mentioned in the current draft so we clarified this in the caption of Fig. 4b.
>
> - **"How is the Eikonal constraint satisfied from the ResUnet++ prediction?"**
>
>   In our setup, the Eikonal constraint is learned from the ground truth, which is constructed to satisfy this property. Note that this is because we use Danielsson's algorithm for converting binary masks of layer boundary into signed distances which computes the Euclidean distance from each pixel to the nearest object boundary, satisfying the Eikonal equation by maintaining a gradient magnitude of one in proximity to boundaries.
>
>   To improve the clarity, we have added further details about satisfying the Eikonal constraint in Appendix A.1.
>
> - **"Figure 1 seems to be describing the idea of SDF and isosurface extraction. A more useful diagram may have been a schematic of how the probabilistic SDF model was implemented (2D image - ResUnet++ - pixel-wise prediction of mean & std of SDF) vs. regression or other methods."**
>
>   In response to your feedback, we have included a schematic (newly added Fig.5) for all four methods (REGR, pREGR, SDF, pSDF) in the appendix.

---

### Author Response · Authors · 2024-03-16
**Rebuttal Summary**

We extend our sincere gratitude to the reviewers for providing detailed feedback on our manuscript. Following the insights and suggestions offered, we have made enhancements to our manuscript, refining both the presentation and the empirical substantiation of our work. Notably, we have clarified the novelty of our approach towards the end of the introduction, ensuring our contributions are well highlighted. The updated manuscript now includes detailed experimentation with a pixel-wise segmentation model, offering a comparative analysis that showcases the superiority of our method. This analysis is further expanded with experiments comparing our approach against Monte Carlo Dropout and Deep Ensemble methods, illustrating the effectiveness of our measure of uncertainty. We have also elaborated on satisfying the Eikonal constraint. Additionally, inclusion of Fig. 5-7 (along with analysis) now offers more insight into the challenges addressed by our method and the learning pipeline used in our approach.

We believe that the improvements considerably enhance the clarity and coherence of our work, effectively addressing all concerns raised by the reviewers.

---

### Meta-Review · Area_Chair_PqwN · 2024-04-04

**Recommendation:** Accept (Poster)
**Confidence:** 4

**Metareview:**

The paper proposes a method for retinal layer segmentation in Optical Coherence Tomography (OCT) scans using probabilistic Signed Distance Functions (SDF) predicted by a ResUNet++ model. While the approach shows promise and demonstrates performance improvements over baseline methods, reviewers raise concerns about the novelty of the method, the clarity of certain visualizations, and the depth of comparison with existing methods.

The paper presents a well-structured and promising approach for retinal layer segmentation in OCT scans. To strengthen the findings, reviewers recommend a more comprehensive comparison with existing segmentation methods, clearer visualization of uncertainty estimates, and a more thorough literature review to demonstrate the novelty and effectiveness of the proposed method. The authors have addressed many of the concerns.

---

### Decision · Program_Chairs · 2024-04-06

Accept (Poster)